# Failed Emancipations: Youth Transitions, Migration and the Future in Morocco

Carles Feixa [1], José Sánchez-García [1,*] , Celia Premat [2] and Nele Hansen [1]

1 Communication Deparment, Pompeu Fabra University, 08018 Barcelona, Spain
2 Social and Cultural Anthropology Department, University of Barcelona, 08001 Barcelona, Spain
* Correspondence: jose.sanchez@upf.edu

**Abstract:** Several authors have highlighted the importance of marriage as a social marker that alters the social categorization of individuals and their relationships from youth to adulthood, according to the cultural construction of the life course in Arab countries. This article analyzes the interaction between the socio-political framework (structure) and the capacity for individual action (agency) in the context of biographical experiences for achieving emancipation in Morocco. This perspective responds to different authors' demands to include young people's subjective approaches in the analysis process. This study is guided by the following questions: What capacity do young Arabs have to decide the orientation of their life trajectories? Which factors (cultural, family, socioeconomic, educational, etc.) generate young people's expectations regarding their transition to adult life? What are the social constrictions that lead to failure in the emancipation process, according to Arab societies?

**Keywords:** transition; emancipation; youth; Morocco





> *Blind and dirty, asked me for a dime, a dime for a cup of coffee.*
> *I got no dime but I got some time to hear your story.*
> *(Wharf Rat, Robert Hunter, 1969)*

## 1. Introduction

In Morocco, the rapid demographic change in recent decades has facilitated the emergence of a youthful population that is producing a significant change in the social composition of the country. According to data obtained during the survey on the socio-economic situation of the child-and-youth population (15–29 years old) carried out within the framework of the Sahwa project, this age cohort represents 32.10% of the population, consisting of 16.14% boys and 16.09% girls. This demography implies that young people are occupying an increasingly central place in the public sphere as a result of a combination of the neoliberal globalization process, the emergence of a civilizational discourse in which Islam positions itself in opposition to the West, and increasing levels of college graduates who are unemployed [1]. Young people feel trapped in a world in which they are required to be married to become adults after achieving economic autonomy. As a consequence, young people—both male and female—are placed in a social structure that understands the youth stage as a 'transitional period' to adulthood, culminating in marriage. The alternative is to become marginalized in their attempts to reclaim their youthfulness [2].

This article aims to summarize the initial research findings on youth, agency and migration. After establishing a theoretical framework on emancipation, the transition to adult life and youth agency in MENA countries, we present a first ethnographic analysis of the data obtained during field work carried out in the cities of Salé and Tangier, Morocco, in 2021, describing some of the elements that frustrate the emancipation of young people in a situation of exclusion and marginalization.

Observing the impediments to emancipation makes it possible to understand one of the main effects of this failed transition: that young people want to migrate to Europe because one of the main driving forces of migration in Morocco is the lack of expectations

for the future. Forced into a nomadic and street life, these young people are located in a permanent state of adolescence. They seek and obtain their vital resources in an informal way, which places them in a vulnerable position. This study responds to different authors' demands to include young people's subjective approaches in analyses and explore the motivations for delayed emancipation [3].

## 2. Methodological Questions

We understand methodology as a way of organizing research "with the purpose of solving or dealing with social problems and, simultaneously, with related scientific problems, differentiating and integrating knowledge from various disciplines of scientific and social knowledge" [4]. In this case, ethnography is a key artifact for obtaining necessary information. For us, ethnography is an open and dialogical mode of social research, which resists easy codification and allows the construction of knowledge in a collaborative way among all the participants in a study. We propose, then, a relational ethnography as an alternative form of traditional fieldwork based on groups and locations, to focus on processes that involve configurations of relationships between different agents or institutions [5]. The focus of fieldwork is therefore to describe a system of relations, "to show how things hang together in a web of mutual influence or support or interdependence or what have you, to describe the connections between the specifics the ethnographer knows by virtue of being there" [6]. Thus, rather than to construct "subjects", in an inductive or deductive way, the objective is to observe configurations of relationships. The methodology constructs the pitch, which in our case is the "youth migrant micro-cosmos" encompassing the following agents: the State, academia, media, gangs, and young people themselves, among others. The aim is to understand how this field works and which position each of these agents occupies (although positions are variable) as well as to determine which dynamics are generated.

To achieve our purposes, two locations were established as analysis units: Salé and Tangier. During the fieldwork, we held six focus groups and five narrative interviews with young people, as well as group interviews with the main relevant NGOs that work in the cities. Rather than researching migratory processes and trajectories, we interpreted processes with blurred limits. Social workers and other agents related to the migratory process, as well as young people themselves, participate in this system of relations. This style of ethnographic practice provides us with multiple forms of data, a dialogical relationship between researcher and participants and the production of knowledge that potentially contributes to social reform. Therefore, we used an interview technique in which conversation is combined with walking and making observations in the local area in which the people move when carrying out their activities or practices, that is, the "action" of young people in their space. In this method, the informant leads the way and decides where to take the researcher. This makes it possible to carry out specific fieldwork in the different spaces of the informants' worlds, and the informant plays a much more active role than in the traditional interview situation.

Finally, ethically, young people were the main protagonists during the research process. This perspective corresponds to the general concern among ethnographers that research should not use informants as mere sources of information but that, rather, from an ethical perspective, the research should serve the interests of those who agree to participate in it. The identity and confidentiality of the participants was preserved by using pseudonyms in the text and saving the data in a secure server.

## 3. Youth Agency and Emancipation in the MENA Region

It is well known that young people's ability to decide their own life path is conditioned by socially imposed identities. Mary Douglas [7] discovered that social categories are defined by attributes and characteristics sustained through collective representations created by social institutions, such as the family or the peer group. Therefore, the construction of the category of "youth" hides intersectional identities imposed from hegemonic spheres. The intersectional mechanism determines forms of social identity from which individuals

design their place in the social sphere according to their imposed identity attributes. For young people, these mechanisms create disadvantaged or privileged circumstances in which they decide on their life course [8].

To understand how youth agency is constrained, a good starting point is a critical sociological analysis of young people's relationship with social institutions, understanding agency as the ability of individuals to act in the social space in which they are involved. Therefore, young people are agents who seek to negotiate their lives within the reality that surrounds them, producing a transformation within it as well as transforming their own reality [9]. However, young people do not constitute a more socially homogeneous group than adults. Thus, young people consider what is expected of them in each social situation but intentionally manipulate their situations to follow their individual strategic interests. This allows them to intervene in the structure and change it. However, this path is always modified by the experience of gender, class and "race". Consequently, in this article, we propose a contextual approach that provides an insight into how different groups of young people in Morocco situate themselves in their local contexts and which barriers prevent their emancipation. Just as importantly, the approach offers the possibility of involving young people in understanding the social practices in which they are engaged, including the institutionalized limits of collective and individual agency, as well as the potential to be a part of the wider challenges to those social structures that perpetuate inequality and social injustice [10].

Identities determined through the intersectional mechanism are superimposed on the elective identities of young people, confirming privileged or unfavorable positions in the social structure [11]. These positions cause significant differences in life trajectories and transitional turning points, such as marriage in Maghreb societies [12]. Thus, the gender, social class, household, cultural capital and family capital that individuals possess, or which is attributed to them, conditions the decisions of young Moroccan people on their way to adulthood. The hierarchization of these social conditions and identities confirms that the social inequalities imposed by attributes associated with certain identities become a matrix of domination defined by understanding these identities as "vectors of oppression and privilege" [11]. Thus, the interest in the notion of youth agency can be said to reflect two different approaches: the first considers young people as a potential danger to the social order; while the second considers young people as social agents potentially involved in processes of social and cultural innovation and as producers and distributors who generate important social benefits.

In the MENA region, the categorization of a person as a "youth" leads to blurred frontiers. As a Moroccan stakeholder remarks: "many countries have different definitions: for Morocco, a young person is from 18 to 30 years old; in Egypt, from 18 to 35, and in Bahrain, from 18 to 40. The definition depends on when *the person leaves home and becomes the head of the household*. Thus, a person with a household becomes a person with a position in society, and so defining exactly who is a youth is very complicated" (Moroccan social worker). Although the starting point of the period is a defined limit, the decisive attribute for young men is to leave the parental house and become the head of their own household. Therefore, obtaining a job is very important for achieving economic independence and financing their marriage [13]. Thus, the "youth issue" in the region is often expressed, paradoxically, as both a problem and an opportunity. As a problem, it is related to security, anxiety about the increase in the youth population in an adult-centric society, unemployment, inequality, drug use, extremism and being victims and, sometimes, perpetrators of structural violence. As an opportunity, the "youth issue" means that young people are perceived as a fountain of wealth for the country due to their work capacity and their desire to migrate and send back resources to their families [14].

## 4. Being Young in Morocco

In the Moroccan context, hegemonic discourses about young people provoke stigmas that hinder their emancipation, especially concerning "street" and "deviant" young peo-

ple. Being young in Morocco means managing a wide range of complex identities and attempting to navigate social circumstances that impose identifications on young people themselves: peer recognition is often more important than parental recognition; the feeling of personal freedom coexists with the awareness of social control and the relationship with hegemonic discourses is diverse [15]. However, the patterns and values of social institutions (religion, kinship, gender, political and economic structures) related to young people are not changing as rapidly as young people themselves; thus, young people are placed in contradictory situations. On one hand, Islam and the family form a discursive group and, on the other, consumption as an element of identity is propitiated by forms of neoliberal capitalism [16].

These two vital orientations constitute hegemonic, dominant and adult-centric ways of understanding the social category, "young", which, on many occasions, relegates young people to the social margins [2]. These schemes are external and superior to everyday experience; they form guides for life and are characterized by ambiguity and polysemy. These guides have two relational dimensions: with daily concerns and experiences and with other vital models. In short, they are guides that promise to give meaning and direction to individuals and their everyday experiences. In the MENA region, the family and Islam, on one hand, and neoliberal capitalism on the other, develop together and influence each other, implying two attitudes towards life: capitalism, with an emphasis on success achieved through profit and consumption; and the Islamic family, focused on morality, the future and "eternal" rewards. However, the promises of both are transitory: that of capitalism is literally consumed by the difficulty of fulfilling it due to economic shortages, while the awareness of future religious rewards leaves individuals in a constant state of tension between the two objectives [16]. As a consequence, the youth cultures that migrant minors and young people build continually move between these two often contradictory worlds

In this general framework, as stated by the young people and minors interviewed in Tangier, two concepts are essential for understanding the dialectic between marginalization and life expectancies among Moroccan youth groups: hogra and karama. Hogra, literally 'humiliation', refers to any situation in which one individual humiliates another for various reasons. This includes social, family, economic, political, cultural or identity humiliations. The minors and young people interviewed highlight situations in which they suffered in different social institutions and with different agents. For example, when a young man tried to sell tissues on a bus, a traveler mocked him and, slapping him, forced him to get off the bus. These humiliations are common for lower-class youth in Morocco and, for those who live on the streets, they are aggravated due to their position in the social structure. This prevents them from carrying out their plans for the future, frustrating their emancipation and life plans. For many of the young people interviewed, these humiliations are caused by the dorof, their structural position in the social system:

> "All of us who have a dorof of poverty have the idea of migrating; on the other hand, young people who have a stable economic situation have no reason to leave the country." (Street Boy, 14 years old)

At this point, it is important to remember that young people have a right to live their youth with hope for the future. Facing their social, educational and cultural situation, young people seek the karama, literally 'dignity', with which to live their youth and, in particular, to carry out their emancipatory life projects [17]. At the same time, the changes in the structural conditions of socioeconomic insecurity have a significant impact on their aspirations, expectations and opportunities to plan future trajectories, creating various situations of disorientation and difficulty in their attempts to solve their problems.

*Tangier Stories*

In recent decades, Tangier, a border city in the north of Morocco and a meeting point between the Mediterranean and the Atlantic Ocean, has become a transition zone towards Europe for both young Moroccans and sub-Saharans. Since the 1960s, there has been a strong wave of migrants from rural areas attracted by the industrial development of the

city. In recent years, the development of infrastructure, real estate, tourism and industrial projects has inserted Tangier and its region into the flows of the globalized economy. The neoliberal policy carried out by the Alaouite State in recent years aims to consolidate the attraction of international financial capital, Arab, European and North American economic actors and migrant members of global classes, that is, the groups of people who travel throughout the planet directing the global economy [18,19]. Although this expansion generates greater economic growth and strong competitiveness in the international arena, the social cost in terms of spatial destructuring, as well as inequality and precarious working and housing conditions for the poorer social strata, remains high. In addition, as a consequence of the permanent permeability of the border with Europe, especially the departures and arrivals of migrants residing in European countries, Tangier is increasingly linked to transnational spaces [20].

In this context, we met minors and young people who wanted to escape and build life projects. Expelled from Morocco due to the serious socioeconomic and political situation, they see no possibility of developing their lives in Moroccan territory. Thus, migrant youths and minors are subject to two dynamics that act at different levels, but converge at their origin. The first is an intervention "from above" on migration, whose initiative comes from the rulers and from contemporary geopolitics, with a neoliberal vision that is very open to privatization and the call of the international finance capital of the global classes. The second refers to a movement "from below", marked by economic transformations in the area and new opportunities in terms of investment, employment and exchange, as well as the transnational mobility of Moroccan migrants themselves, who, across the borders, create their own life, work, social and labor network spaces, their own transnational wasta'. Generally, minors and young people, in this desire to migrate, are unaware of the vital dimension of the changes they seek in their lives, as they do not understand its context or meaning.

In Tangier, we contacted two groups of young people and minors in a street situation, with whom we visited the places where they spent their days and nights. The groups were separate, although they had relationships with each other. The first group lives under the recently built bridges of Boulevard Mohammed VI. It is made up of young people, both girls and boys, between 12 and 21 years old. Despite living on the street, these young people maintain relationships with their families if they live in the city. This is the case of Amina, who has three brothers and remains in contact with her mother, who lost her job in the Spanish enclave of Ceuta as a result of the pandemic, an example of the permeability of the border and the importance of entry and exit for the economy of the less favored classes in Tangier. Her mother is aware of her daughter's situation; she is in favor of her daughter migrating and even wants to finance the illegal trip (which costs between EUR 3000 and 5000 to board a boat for a journey of about 20 min to an Andalusian beach) because it would help her seek her happiness. However, the most controversial issue is that Amina prefers to live on the street rather than with her family, which has caused conflict with her mother.

Another situation is that of Mariam, who arrived in Tangier fleeing from her mother (who had been abandoned by her husband) and who has suffered from conflicts and mistreatment. She left the family home together with a brother and a sister, now pregnant: "I left home after a fight with my mother and we all left" (Mariam, 17 years old). Finally, Zeynab is a 21-year-old girl who has taken on the role of the mother of the group, preparing food, caring for and protecting them. However, at the same time, Zeynab has adopted bodily forms that are typical of masculinity, understood through normative notions of gender, in relation to this function. This is why Zeynab dresses like a boy, has short hair cut like a boy and adopts the forms of corporality and movement typical of boys in order to demonstrate strength and a predisposition towards using violence to protect the group.

Another of the groups we came into contact with was made up of young people, both girls and boys, between the ages of 14 and 18, who live together in a plastic shack next to the railway wall that divides the city into two zones. When they arrived at their place of residence, they were eager to communicate their experiences, especially the needs they had. When the group's reference educator arrived, the children ran to him and

he was overwhelmed by their explanations of their needs, many of them basic, such as food, clothing and, above all, health care, which is why he always carries products in his backpack, including disinfectants, anti-inflammatories, and band-aids. This was the intervention he carried out in the street with these children. They were given a band-aid instead of the comprehensive interventions necessary to ensure their right to housing and education, but also to adequate food for their life stage and, ultimately, their right to a full life. In addition, the physical degradation to which they are subjected by life on the street makes them pessimistic about the possibility of traveling to Spain, due to the lack of family support and their mental and physical health problems. Nur and Ahmed had not been home for two years; they kept in touch with their mothers, but not with their fathers. Tawfiq's parents are divorced and remarried; she has seven siblings. Half of her siblings live with her mother and the other half with her father. Two of her brothers are married, her mother works in a flea market in the center of Tangier near a mosque and sees her regularly, but her drug problems led to her being kicked out of her home for endangering the family honor with her behavior. Osama managed to enter Ceuta when he was twelve years old, wandering and living on the street without obtaining a place in the juvenile center. Finally, he decided to return because he was afraid; it was the first time that he had been separated from his mother. For him, returning to Morocco was the worst decision of his life. Marwan does not keep in touch with his mother, but he talks to his father from time to time. With five siblings, only his mother works, in a bakery, which provides insufficient money for the family's needs. For this reason, he would like to migrate, to be able to help his mother and achieve his emancipation, which has been frustrated in Tangier. These youngsters have developed a critical discourse about the family. In their opinion, their education was inadequate and they emphasized the importance of the peer group to gain life experience and develop their personality and social skills towards maturity. In this regard, Marwan believes that:

> Friends can teach you positive things, because when you are in a strict family, trapped between four walls, and then you get out, you are free! You don't know what to do! But when you are in the world you come to learn what to do and what to avoid, you will also learn spontaneously with the people you go out with . . . some friends have taught me things that I did not learn from my family. Prayer, for example. My parents haven't told me to pray since I was six years old.

The personal degradation as a consequence of living in the streets made the youths' journey to Europe almost impossible. As Amina remarked,

> I don't feel like migrating anymore, I've failed so many times that I can't any more . . . in order to be able to migrate you don't have to look back, don't think about anything or anyone. Knowing that you are leaving a part of your life in one place and that you will leave forever leaving everything behind . . . today migrating is for the lucky ones . . . I've already lost hope, I don't think about anything anymore (ana fkads el amaal, me knfakar fi tahaja)

These youngsters were expelled to the streets, without rights, land searched every day for the means to survive, using cola or other types of drugs to get through each day; they were degraded until they lost their physical and mental health and self-esteem, continuously supporting hogra forms.

## 5. Family and Emancipation

The family is a source of social capital as it is one of the basic pillars of MENA societies and key to achieving emancipation and adulthood. This means that family honor, respectability, or *wasta'* and *marifa'* connections, are assets for young people to reach emancipation through marriage. All these attributes, which are related to education, employment and economic circumstances, make up the prestige and reputation of minors and young people. The family also imposes *dorof* on its members. These conditions influence the possibilities and opportunities of young people and the development of their capacities.

This importance of kinship for the social life of the young people explain why, during the conversations, seminars and collective interviews carried out, family relationships repeatedly appeared as a significant element for understanding motivations and also the form that the migratory process adopts. In some cases, encouraged by the families' own socioeconomic situation (which can be described as a family project), migration represented a significant socioeconomic improvement.

In other cases, migration was motivated by expulsion from the family itself due to situations related to vulnerabilities, lack of protection and, in some cases, mistreatment. As Nur explained, "me and my sister ended up on the street . . . because our fear of our father. We worked in a clothing factory and one day we didn't go . . . The fear we had of thinking about how my father would take it when we found out that we went to work could become so terrible that we decided not to go, we knew that if he found out it would hurt us a lot, and it could have very serious consequences." Therefore, it is the family that becomes the key element in understanding the different realities of potential migrants, even determining their chances of success. Nevertheless, in both models, youngsters are searching for a way to emancipate themselves and follow their aspirations and motivations.

A good example is Fatima, abandoned and mistreated by her husband, with seven children; she sells flowers and paper tissues in the street. Her aim is for her children migrate; she encourages them and helps them as much as possible in this regard. For one of her children to reach Europe would imply an improvement her conditions in Tangier. In fact, to facilitate the trip, she moved from Taza (wilaya of Fez-Mequinez) to Tangier because her children ran away from home to reach the northern city and try to migrate. In this way, she prevented her children from becoming street children, although this has itself resulted in hardships, including the drug addiction of one child. One of her children has lived in Melilla for two years. He has reached Melilla five times since he was seven years old. The first time he hid under a trailer, the second time he hid in the engine of a car and, the last time, he sneaked through the military borders between Morocco and Melilla through a hole that only a few people knew about and of which the military had no knowledge. The fourth time, he entered as a merchandise vendor with his brother and, the last time, he swam. Fatima is happy because her son is in a good condition. He is now trying to reach the peninsula and to take his mother and his brothers and sisters with him so that they can live in better conditions. These desires to migrate are not limited to the boys; Fatima also wants her daughters to migrate because "they do not want to live in a country that sees how their mother and their brothers and sisters suffer every day" (Fatima, mother, 48 years old).

To a greater or lesser extent, many young people, especially women, consider that the family continues to provide adequate education and a good path towards emancipation. Growing up in a "respectable" family, that is, honest and educated, provides them with a safe anchor and an environment of support until they achieve their aspirations, always passing through the initiation that marriage entails. For these young women, the family continues to be a point of reference and support and the source of the most respectable and socially esteemed values: a highway to adulthood directed by the dominant transition model.

From a contemporary, modern and secular point of view, the family can be thought of as a model of a "totalitarian" group, a pressure group, but it is also an environment of education, support and reinforcement. The young people of Tangier show a strong attachment to their family (due to both positive and negative experiences), which continues to be their emotional and social point of reference and their economic support, in the absence of a State or national social-service support system. When the family fails, together with the economic precarity and perceptions of deviancy according to dominant discourses in the MENA region, the process of transition to adulthood and emancipation is almost impossible. The solution is to try to cross to Europe to achieve their life plans and aspirations.

## 6. Conclusions

In the current situation, while some young people experiment with new perspectives and cultural practices about what youth should "mean" in terms of space and personal

autonomy, others prefer the idea of completing their transition quickly, according to the dominant life stages. However, some cannot achieve any type of emancipation, as they live on the street and lack the necessary resources for building life plans. Nevertheless, in all cases, the central issue is choice: choosing to participate in the traditional family unit and/or choosing a more individualistic worldview centered on independence and the right to define a life project. Thus, the planning of their own lives, the transition to adulthood and emancipation is not exclusively in the hands of the young people themselves in Arab society, as it is marked by adult-centric and patriarchal control of the lives of young people.

In general, we can establish that young people who adhere to figurative cultures have an easier path to emancipation than young people who try to follow a prefigurative orientation in their decisions. Consequently, reaching adulthood and social maturity depends on the performative agency of individuals. However, the young people of the MENA region find themselves in an agency framework limited by social restrictions: a "bounded agency" [21]. In short, the youth of the MENA region are simultaneously trapped in structural realities constrained by the cultures of societies that are likely to judge many of their attitudes and practices as "deviant". These realities impose barriers to youth aspirations as they escape social control and also to the socially consecrated road to emancipation.

**Author Contributions:** Conceptualization, J.S.-G. and C.F.; methodology, J.S.-G.; software, N.H.; validation, C.F., J.S.-G., C.P. and N.H.; J.S.-G. and C.P. formal analyses; investigation, J.S.-G., C.P. and N.H.; resources, N.H.; data curation, J.S.-G. and C.P.; writing—original draft preparation, J.S.-G.; writing—review and editing, J.S.-G.; visualization, N.H.; supervision, C.F.; project administration; funding acquisition, C.F. All authors have read and agreed to the published version of the manuscript.

**Funding:** This paper is part of the collaboration agreement between the Casal d'Infants del Raval (RASSIF project, financed by the Catalan Agency of International Cooperation to Development) and the TRANSGANG project: Transnational Gangs as Agents of Mediation: Experiences of conflict resolution in youth street organizations in Southern Europe, North Africa and the Americas, financed by the HORIZON-2020 Program, European Research Council—Advanced Grant (H2020-ERC-AdG-742705). PI: C. Feixa. 2018–2023. Pompeu Fabra University, Barcelona.

**Institutional Review Board Statement:** The study was conducted in accordance with the Declaration of Helsinki, and approved by the Institutional Review Board (or Ethics Committee) of POMPEU FABRA UNIVERSITY (protocol reg. nº 0046, 19 April 2017).

**Informed Consent Statement:** Consent was obtained from all subjects involved in the study.

**Data Availability Statement:** Not applicable.

**Conflicts of Interest:** The authors declare no conflict of interest.

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
