# Peer review of "Failed Emancipations: Youth Transitions, Migration and the Future in Morocco"

_societies, doi:10.3390/soc12060159_

Round 1

Reviewer 1 Report

On the face of it, there is much of merit here and I would have liked to be more positive about it. It is an important topic and there is some relevant engagement with sophisticated literatures and argument. The fieldwork element was potentially exciting and important. However, as it stands, the paper has a number of substantial weaknesses which need addressing before it could be published.

1. The analytical framework references some important conceptual keys: the social construction of youth as a category, how this is externally produced by socialisation institutions and processes as well as young people's own identity formation responses; intersectionality; the processes and impacts of dual narratives of youth as threat/hope for the future; adulthood as emancipation. However, none of these are fully explored and, more importantly, the relationality between them is muddled and inconsistent. It is never clear exactly what the research question is, or how these conceptual tools will be used to answer them.

2. The methodological rationale for the fieldwork is not explained, nor is there any detail about how or when it was done, by whom, and with what ethical protocols (which is essential when dealing with minors and vulnerable subjects as is clearly the case here). Ultimately the primary data amounts to very brief biographies of six young people, which are not explored fully against criteria or categories drawn from the theoretical material at the start. Given the scale of the assertions about 'neo-liberalism', Islam, consumption culture, etc. the data is not enough to really pull all this together (even though there are specific insights which I would completely agree with). In sum, this is all very subjectively constructed and lacks methodological rigour. This is a real shame: access to such marginalised and vulnerable groups as discussed here is massively important and it is to the author(s) credit that the effort has been made. But so much more was needed, including acknowledgement of the challenges of conducting biographical research in such settings and more detail to round out the stories and make them more than selective sound bites. 

3. The English language really lets the paper down. At times it actually doesn't make sense, although this may in part be because of the desire to make grand, albeit unsupported, statements like "it is important to remember that being young is a right that must be lived with a future perspective.....".

4. Overall, the paper tries to do too much with too little. The value of the paper comes from the potential (but in this instance insufficient) treasure trove of empirical data. Exploring the narratives in much greater detail would allow the researcher to trace more convincingly the intersections of structure and agency as they a) allow the young people in question to construct their identities both autonomously and relationally; b) understand how their life trajectories have been/will continue to be shaped by structural conditions and c) identify and assess when and how the young people have exercised agency (perhaps better understood as resistance - see King) and d) evaluate how and when this all combines to amount to or impede 'emancipation'.

In sum, lots of good ideas and intentions, and potentially great empirical data, but really needs to be properly structured as both a research project and a piece of writing.

Author Response

Dear Reviewer,

Thank you very much for your useful commentaries and suggestions. According to we have included a methodological section to clarify the way in which the research has been carried out, including the techniques used and the ethical approach adopted.

"Overall, the paper tries to do too much with too little", the claims of the article have been lowered and identified as a first presentation of data that we continue to analyze in the framework of the TRANSGANG and RASSIF projects, and that in the coming months will bear new fruit from this first approach. 

About your main points just to clarify:

  1. "However, none of these are fully explored and, more importantly, the relationality between them is muddled and inconsistent. It is never clear exactly what the research question is, or how these conceptual tools will be used to answer them". It is impossible to fully explore in a short paper like this. Anyway in the case of social construction of youth category in MENA countries and socialisation you can find several references of the author's establishing this preliminary framework on MENA youth.
  2. "Given the scale of the assertions about 'neo-liberalism', Islam, consumption culture, etc. the data is not enough to really pull all this together (even though there are specific insights which I would completely agree with)" All of this is a general framework well established in field of Muslim youth studies and it is not necessary to prove it again. 
  3. "including acknowledgement of the challenges of conducting biographical research in such settings and more detail to round out the stories and make them more than selective sound bites". Totally agree but this is another paper and it is not in the aim of this article that it is a first findings analyses. Moreover, we need to publish the data in project reports.

  4. "a) allow the young people in question to construct their identities both autonomously and relationally; b) understand how their life trajectories have been/will continue to be shaped by structural conditions and c) identify and assess when and how the young people have exercised agency (perhaps better understood as resistance - see King) and d) evaluate how and when this all combines to amount to or impede 'emancipation'." Absolutely agree but we think that we can do it with more accurate analyses in next articles.

Reviewer 2 Report

The work presented is of great interest. It addresses a very relevant issue: the situation of migrant boys and girls. A group which often suffers from social stigmatisation that is detrimental to the rights of these minors. in general terms, I consider the subject matter of the manuscript to be very interesting for the journal. In relation to the methodological, the article has the good sense to carry out a qualitative study.

in short, the article presented is necessary in the social reality analysed. The authors are congratulated for this.

Author Response

Thank you very much for your review. All the suggestions has been improved in the second version of the paper.

Round 2

Reviewer 1 Report

I appreciate that a discussion of the methods used has been included in the revised text and also that the ambition of the paper is to 'provide an ethnographic first analysis of the data' of the project at hand, and not a complete final analysis. Nonetheless, the revised document addressed almost none of the substantive concerns raised previously. 

Sections 3 and 4 introduced a number of admittedly important considerations in research on, and understandings of, youth, but these remain ontologically unlocated. Moreover, whilst it is claimed that they present 'a general framework', what they actually do is make a number of assertions, the relationality of which is not established and which are not followed through in a clearly identifiable way in the subsequent very short empirical discussion. This is very frustrating because it is clear the authors are well-read and - if the dots were joined properly - this could be really exciting work. The authors indicate that much is left to be said in later publications but, if all the substantive material of the empirical research (because what is here is very limited) and the significant analysis of it within a fully-developed framework which shapes not only the analysis but the methodology by which it is retrieved, then what is the point of this 'taster' article? There is not enough empirical material to really illustrate or unpack the larger 'framework' claims and the framework is not demonstrably used to shape either the data collection or the subsequent arguments such as they are. So just saying that everything has either already been done elsewhere or will be left for future publications, doesn't resolve the inherent limitations of this particular piece. 

I think the authors are doing themselves (and their research subjects) an injustice because they are seeking to publish what is ultimately an enormously promising piece of research prematurely. The knowledge and understanding of the literature is here, but it is not being used to great effect. Equally, the empirical research is important and promising but ethnographic research cannot be properly presented in such brief vignettes and if, as the authors claim, the participants are not to just be sources of information, we need to hear THEIR voices more clearly in this and know much more about their lived experiences and the meanings THEY attribute to it.

Author Response

Dear Reviewer,

We are improve the written language and introduce new section on methodology. 

Best